# BIOS-Based Server Intelligent Optimization

**DOI:** 10.3390/s22186730

**Published:** 2022-09-06

**Authors:** Xianxian Qi, Jianfeng Yang, Yiyang Zhang, Baonan Xiao

**Affiliations:** School of Electronic Information, Wuhan University, Wuhan 430072, China

**Keywords:** reinforcement learning, BIOS, server, performance optimization

## Abstract

Servers are the infrastructure of enterprise applications, and improving server performance under fixed hardware resources is an important issue. Conducting performance tuning at the application layer is common, but it is not systematic and requires prior knowledge of the running application. Some works performed tuning by dynamically adjusting the hardware prefetching configuration with a predictive model. Similarly, we design a BIOS (Basic Input/Output System)-based dynamic tuning framework for a Taishan 2280 server, including dynamic identification and static optimization. We simulate five workload scenarios (CPU-instance, etc.) with benchmark tools and perform scenario recognition dynamically with performance monitor counters (PMCs). The adjustable configurations provided by Kunpeng processing reach $2^{N}(N>100)$. Therefore, we propose a joint BIOS optimization algorithm using a deep Q-network. Configuration optimization is modeled as a Markov decision process starting from a feasible solution and optimizing gradually. To improve the continuous optimization capabilities, the neighborhood search method of state machine control is added. To assess its performance, we compare our algorithm with the genetic algorithm and particle swarm optimization. Our algorithm shows that it can also improve performance up to 1.10× compared to experience configuration and perform better in reducing the probability of server downtime. The dynamic tuning framework in this paper is extensible, can be trained to adapt to different scenarios, and is more suitable for servers with many adjustable configurations. Compared with the heuristic intelligent search algorithm, the proposed joint BIOS optimization algorithm can generate fewer infeasible solutions and is not easily disturbed by initialization.

## 1. Introduction

Driven by the strong demand of the internet industry, more traditional industries have undergone digital transformation, and servers are also widely used by enterprises to build network applications. For server buyers, high-performance servers can provide their users with a better experience and meet business needs. However, the performance of the server mainly depends on its hardware configuration. The better the configuration is, the higher the performance, and the corresponding cost will also increase. Therefore, it is necessary to improve server performance based on inherent hardware devices. Server users can conduct performance tuning at the operating system layer or at the application layer according to the specific application. For example, Li et al. implemented dynamic performance optimization of the Apache Web Server by tuning some important parameters, such as ‘MaxClients’ and ‘KeepAlive’ [1]. For database application servers, SQL queries always have an optimization space. Reference [2] investigated a series of query optimization techniques for improving the energy efficiency of relational databases and NoSQL databases.

Another effective technique is to improve the performance of a single server through a reasonable hardware configuration. Reference [3] showed that hardware data prefetching could improve server application performance, but it did not provide a detailed description of how to configure it. References [4,5,6] produced a remarkable performance improvement by adjusting the hardware configuration on Intel and POWER8 servers. Nevertheless, these are all hardware prefetching configurations. On the Taishan 2280 server, the Kunpeng 920 processor [7] also provides a large number of adjustable configurations, such as L3 cache prefetch configurations and HHA (HCCS Home Agent) [7] related configurations. The Basic Input/Output System (BIOS) [8] on the server can control the underlying registers to set these configurations. For example, the L3T_PREFECTH register can control the “prefetch_utl_ddr_en” function, which represents “whether to allow automatic threshold reduction according to the utilization of DDR”. For more registers and configurations, please refer to [9]. The configurable register bits can be regarded as BIOS switch control items. Reasonable configuration can improve the performance of a multicore server system. Since there are only four configuration items [4,5], exhaustive search methods were used to find the best configuration for a specific application. Reference [6] considered the configuration optimization of 25 bits, but they only filtered out the parts that had substantial performance gains and did not fully consider the possible nonlinear relationship between configurations. In this paper, the number of BIOS configuration items is larger, and it is impossible to manually select a configuration representing near-optimal performance out of such a huge search space. Therefore, it is necessary to resort to intelligent optimization algorithms. Evolutionary algorithms and local search algorithms based on swarm intelligence are common approaches for solving this type of problem. Genetic algorithm (GA) [10,11], particle swarm optimization (PSO) [12,13,14], whale optimization algorithm (WOA) [15], gray wolf optimization (GWO) [16,17] and other intelligent algorithms are widely used for parameter optimization, and there are versions for discrete parameters. These methods are iterative optimization algorithms that cannot learn the configuration process and may have a higher probability of causing server instability. Since the process of optimizing configuration can be modeled as a sequential decision-making process, we can also adopt reinforcement learning methods. With the development of artificial intelligence technology, deep reinforcement learning has shown great research potential on sequential decision-making problems [18,19].

It can be seen from the literature [4,5,6] that a fixed configuration does not help performance in many cases, and in fact, it may degrade performance due to useless bus bandwidth consumption and cache pollution. Similarly, it is also not appropriate to set a fixed BIOS configuration for the server because servers may run under different workloads, such as CPU-intensive, disk-intensive, memory-intensive, network-intensive, etc. Due to the different performance indicators of various workload scenarios, the optimal hardware configuration is also different. When the server workload is frequently switched, to avoid the trouble of manually checking and adjusting the configuration, this paper proposes a dynamic tuning framework for dynamically adjusting the configuration. First, the workload scenario is dynamically identified through the server performance monitor counter (PMC) data [20], and then, the optimal configuration can be switched for the identified scenario. Server workload monitoring has been discussed in many works [21,22], but these monitoring data come from operating systems, such as CPU utilization and disk read and write rates. Similar to our work, [4,5,6] use PMCs as inputs for machine learning strategies to select the best configuration, but due to the few configuration items, their machine learning strategy directly outputs the optimal configuration, not the workload scenario. When there are many configurations, this method is not suitable. Therefore, this paper uses the workload scenario as the label of the machine learning strategy. BIOS can also collect hardware PMCs by accessing the underlying event register, including cycles, iTLB-loads, branch-load-misses, dTLB-loads, and other event-counting data [23]. To the best of our knowledge, there has been little research on optimizing server performance by dynamically adjusting the BIOS hardware configuration. Therefore, this study makes the following contributions:
(1)To improve the performance of Kunpeng processor-based servers, we proposed a performance tuning framework for dynamically adjusting the BIOS configuration. It monitors the server workload information to identify scenarios and implements performance optimization based on the results of static tuning or empirical configuration.(2)At the static configuration tuning stage, finding a near-optimal BIOS configuration is modeled as starting from a feasible initial configuration and adjusting the BIOS configuration to obtain an improvement. Based on this model, we propose a joint BIOS optimization algorithm using a deep Q-network combining reinforcement learning and nearest neighbor search.(3)With the proposed optimization algorithm, we significantly improve the memory bandwidth rate in memory-intensive scenarios. To further evaluate the proposed static tuning method, we compare it with two metaheuristic methods: genetic algorithm and particle swarm optimization algorithm. The algorithm in this paper is more stable and has a lower probability of server downtime.(4)We have also carried out optimization work in other load scenarios and found that in some scenarios, performance indicators are no longer critical optimization indicators.

The remainder of the paper is organized as follows. Section 2 describes the dynamic-tuning framework. Section 3 and Section 4 discuss methods for load scenario identification and optimization, respectively. Section 5 presents the results of operating scenario recognition and optimization. Finally, this paper is summarized in Section 6.

## 2. Dynamic Tuning Framework

In this section, we describe the dynamic-tuning framework, which is separated into three stages: an offline training stage, an offline static configuration tuning stage, and an online dynamic-tuning stage, as depicted in Figure 1.

*Online dynamic-tuning stage.* This paper applies machine learning to predict server workload scenarios and periodically sets the corresponding optimal hardware configuration during the online dynamic-tuning stage. It collects PMCs periodically and feeds them into a classifier after a fixed time to identify the current workload scenario. Once it recognizes that the scenario changes, the near-optimization configuration is switched in real-time.

*Offline training stage.* The training data for machine learning are prepared offline. We simulate workload scenarios with benchmark software to collect training data in Section 3. Similar to [5], we measure all performance events supported by our processor and perform feature selection. The final dataset is a matrix, and each row consists of performance counter data and scenario labels. The performance counter data are the values of the specific scenario fixed time. Then, we build a classifier using machine learning strategies.

*Offline static configuration tuning stage.* To find better configurations from the inexhaustible configuration, we do not use iterative search methods. Instead, we use reinforcement learning to gradually optimize the initial configuration to find a better configuration. This is the main part of the article. The details are provided in Section 4.

## 3. Workload Scenario Recognition

### 3.1. Scenario Preparation

Scenario identification of the server load is a prerequisite for dynamic optimization. However, it is impossible to simulate all workload scenarios, and the load scenarios running on real servers are relatively fixed. Therefore, in this paper, a coarse-grained division of load scenarios is made. Because of the scalability of the proposed dynamic monitoring algorithm, users can train fine-grained classifiers according to the actual server usage. According to different workload conditions, server workload scenarios can be roughly divided into CPU-intensive scenarios, memory-intensive scenarios, disk-intensive scenarios, network-intensive scenarios, and idle scenarios. The difference in load is mainly due to the different tasks currently running on the server. By running the benchmark tool on the server, various scenarios can be simulated, as seen in Table 1. FIO [24] is a benchmark software used to test hard disk IO, which can stress test hard disks, including sequential read and write and random read and write. IOzone [25] is a file system benchmark tool that provides a variety of IO operations that can comprehensively test the performance of the file system. Sysbench [26] is a multithreading and multifunctional benchmarking tool commonly used to test databases, CPUs, etc. The STREAM [27] benchmark is a comprehensive memory test that is widely used in the testing and server market industries. It supports four operation modes, copy, scale, add, and triad, to test the memory bandwidth performance. Iperf3 [28] is a widely used network bandwidth testing tool that supports IPv4 and IPv6 and can run on Windows, Linux, Android, and other platforms.

### 3.2. Data Processing and Scenario Recognition

A typical multiclass supervised learning problem generally follows four steps: data collection, feature selection, model training, and model evaluation. In various load scenarios simulated by the benchmark test, the BIOS is used to collect PMC data per second. There is redundancy in the original data, and feature correlation analysis can be performed to eliminate some features. The next step is to use traditional machine learning methods for training, such as decision trees (DT) [29], logistic regression (LR) [30], K-nearest neighbors (KNN) [31], and deep neural networks (DNN). In recent studies, DNN can handle complex classification problems well [32,33]. To judge the quality of the final classification model, some common evaluation indicators can be used for evaluation, such as accuracy and precision.

## 4. Workload Scenario Optimization

### 4.1. Markov Model for BIOS Control Optimization

In a specific scenario, the problem of finding a near-optimal BIOS configuration is modeled as starting from a feasible initial configuration and adjusting the BIOS configuration to obtain an improvement. When the BIOS controls the optimization, the state is characterized by the current server’s absolute BIOS configuration and performance evaluation. BIOS configuration can be divided into two categories: integer type, and binary type. There are very few configurations of integer types, so we treat them as multiple binary configuration items by converting them to binary. Therefore, all configurations can be regarded as configuration switch items. The number of BIOS configuration switch items is N (N > 100), and M performance indicators are used for the server’s performance evaluation; thus, the state is represented with an (N + M)-dimensional vector. With the definition of the environment state, the action space can be obtained naturally. Actions can be defined as the flip of N BIOS configuration items and represented by an N-dimensional vector. It shows the BIOS register configuration items that need to be flipped. The position that needs to be flipped is set to 1, and the position that does not need to be flipped is set to 0. It can be seen that the action space is large and reaches $2^{N}$. To narrow the scope of the action space, we make conventions on actions. Given the state s, the optional action can only be to flip a single configuration item. In this way, the configuration optimization problem can be abstracted into multistep tuning rather than one-step tuning. Additionally, the state at the next moment is only related to the current state and the action and has nothing to do with the state of the earlier moment, which satisfies the Markov properties. Therefore, this paper uses the Markov decision process (MDP) [34] to complete the BIOS control optimization model design. The BIOS control optimization model based on MDP is defined as follows:

**Definition** **1.**
*BIOS control optimization is based on MDP (MDP-BIOSCO). An MDP-BIOSCO can be defined as a tuple: MDP−BISOCO=<S,A,P,R,γ>. S is a set of data describing the server state, which is expressed above; A(s) is the restricted set of actions that can be performed in the state s∈S. R is a reward function, and when an action is performed, the server changes from s to $s^{\prime }$. With the state change, the server performance also changes. Based on this change, an instant return r is fed back, and its expected value is $r=E(R(s^{\prime }|s,a))$; γ∈[0,1] is a discount factor to distinguish the importance of future rewards and immediate rewards.*


### 4.2. Deep Q-Network

Deep reinforcement learning (Deep RL) is a combination of deep learning (DL) and reinforcement learning (RL), which is mainly used to deal with high-dimensional states and action spaces [35]. In [36], Mnih et al. proposed a structure named DQN, which could learn to play a range of Atari 2600 video games at a superhuman level. In this study, we employ a DQN because the state-action space is considerably large.

In the high-dimensional state or action space, traditional Q-learning cannot estimate the Q value corresponding to each large state and action space [37]. In DQN, a deep convolutional neural network is used to approximate the optimal action-value function. Nevertheless, when a nonlinear function approximator, such as a neural network, is used to represent the action-value function, RL is unstable or even divergent. To address this problem, Mnih et al. proposed an experienced pool and target network. As illustrated in Figure 2, replay memory is used to store samples $\left(s,a,r,s^{\prime }\right)$ and perform random replay, thereby eliminating serial data correlation. The target Q network is used to assist in calculating the following loss function equation:(1)$$L_{i}(\theta_{i})=E{\left(r+\gamma \;\underset{a^{\prime }}{max}Q(s^{\prime },a^{\prime };\theta_{i}^{-})-Q(s,a;\theta_{i})\right)}^{2}$$
where i is the number of iterations, s and a represent the current state and action, respectively. $s^{\prime }$ and $s^{\prime }$ represent the next state and action, respectively. $\theta_{i}$ are the parameters of the estimation Q-network at iteration i, and $\theta_{i}^{-}$ are the target Q-network parameters at iteration i. $r+\gamma \;\underset{a^{\prime }}{max}Q(s^{\prime },a^{\prime };\theta_{i}^{-})$ is the target value. The parameters $\theta_{i}^{-}$ are updated at every C step from the estimation Q-network.

### 4.3. Joint BIOS Optimization Algorithm Using DQN

To optimize BIOS control in a real server, a joint BIOS optimization algorithm using DQN is proposed based on the above MDP-BIOSCO model. The overall structure is illustrated in Figure 3. The first part involves the training phase and includes three subparts: environment, agent, and experience replay. The second part is designed for continuous optimization during the testing phase.

#### 4.3.1. Environment Design

The environment is the object of agent interaction in a reinforcement learning system. The design of the environment mainly includes two parts: an interactive data record and a function, step().

It takes a few minutes to configure the BIOS and test the performance data on the server. The bios.txt and score.txt files shown in Figure 3 store the BIOS absolute configuration and the corresponding performance score data, respectively.

The core of the environment is the step() function, which contains the contents of the entire environment in Figure 3. The input of the step() function is the action, and the output is the state at the next moment, the reward of the current action, and whether to terminate the training episode. This function directly controls the server to conduct field measurements and calculate the state at the next moment.

The state of the system is obtained at the next moment. The state at the next moment is obtained through end-to-end testing. First, we calculate the absolute BIOS configuration at the next moment from the predicted action at the current time, and then we test the server to obtain the performance evaluation. Finally, the two parts are merged to form a state.An instant reward is obtained. In contrast to the general reinforcement learning task, BIOS control optimization has no specific target, and the desired effect is that the algorithm can obtain better server performance quickly while ensuring the ability to jump out of local optimization. For the STREAM test scenario, the goal is to adjust the configuration, ensuring that the memory scores can increase rapidly and have the ability to find higher scores. Therefore, the reward function is set as follows:

Setting reward, $r_{1}$. The action of BIOS configuration modification causes the performance scores to rise or fall, and $r_{1}$ provides feedback for the current action. There is a large gap in performance evaluation for different servers and operating scenarios. To ensure scalability, this reward adopts the baseline design method. The baseline is the known maximum difference of automated test acquisitions in the initial pre-experimental stage as follows:(2)$$r_{1}=(score_{i+1}-score_{i})/baseline*scale$$

$score_{i}$ and $score_{i+1}$ represent the performance scores of the current moment and the next moment, respectively. Since the score gap may be small compared to the baseline in the actual experiment, we have incorporated a scaling factor scale. By default, scale=100.

Setting a fixed reward, $r_{2}$, when performance scores are higher than the record in this episode. This reward, $r_{2}$, is mainly to drive the algorithm to find higher scores and maintain the ability to jump away from the local optima.

Setting reward, $r_{down}$, when the server is down or works unstably. Taking all of the above into consideration, the instant reward module is described as follows:(3)$$\begin{array}{ll} r & =\{\begin{array}{ll} r_{1}+r_{2} & \;\mathrm{other}\\ r_{down} & \;\mathrm{if}\;\mathrm{server}\;\mathrm{down} \end{array} \end{array}$$

Moreover, the generated samples $\left(s,a,r,s^{\prime }\right)$ are stored in an experience replay memory. These samples are then retrieved randomly from the experience replay and fed into the training process. If it is a downtime sample, repeat the storage k times to increase the proportion of downtime samples.

#### 4.3.2. Agent Decision-Making and Learning

The agent is the main actor in strategy learning. It finds the optimal action strategy through continuous trial and error learning, which mainly includes two aspects: action selection and strategy learning.

Figure 4 is the Q-network structure, the input is the BIOS configuration, and the output is the *Q* value for different actions. For instance, $Q(s,a_{j})$ represents the Q value for selecting the *j*th action at state s. In Section 4.1, we define and restrict actions. Action a means to modify only one configuration, and thus, the output dimension of the Q network is the same as that of the configuration item. At state s, to choose the best action in the set of alternative actions, the decision-maker will use a greedy strategy to take the action with the largest Q value. However, adopting the greedy strategy alone will cause missing the optimal action, so this paper uses the ε greedy policy to ensure a certain probability of exploration. At the same time, to make up for the low intensity initial search problem, a large value of $\varepsilon_{max}$ is set at the beginning of the training of the algorithm, and it decreases according to the decline coefficient $\varepsilon_{decay}$, as the training goes on until it drops to the minimum value, ε as follows:(4)$$\varepsilon =max(\varepsilon *\varepsilon_{decay},\varepsilon_{min})$$

During the policy update phase, the parameters of the Q-network are updated as follows:(5)$$\begin{array}{l} \theta_{i+1}=\theta_{i}+\alpha [r+\gamma \max_{a^{\prime }}Q(s^{\prime },a^{\prime };\theta_{i}^{-})-\\ \;Q(s,a;\theta_{i})]▽Q(s,a;\theta_{i}) \end{array}$$
where i is the number of iterations, s and a represent the current state and action, respectively. $s^{\prime }$ and $a^{\prime }$ represent the next state and action, respectively. $\theta_{i}$ are the parameters of the estimation Q-network at iteration i, and $\theta_{i}^{-}$ are the target Q-network parameters at iteration i. α and γ represent the learning rate and discount factor.

In summary, the BIOS optimization algorithm (PART 1) is applied to the server BIOS control optimization problem, as shown in Algorithm 1.
**Algorithm 1** BIOS optimization algorithm based on DQN (Part 1).**Input**: N Initial capacity of the playback pool,   M Number of final exploration frames for optimization,   T Number of optimization steps per iteration process,   $s_{initial}$ State corresponding to the initial configuration1 Initialize replay memory (D) to capacity N;2 Initialize Q-network with random weights θ;3 Initialize environment;4 For episode = 1, M do5   Initialize environment state $s=s_{initial}$;6   For t = 1, T do7     Calculate Q(s,a;θ) in state s;8     With probability ε select a random action a,       Otherwise, select $a=\arg\max_{a}(Q(s,a;\theta ))$;9     Execute action a in the environment,       obtain a reward r, next state $s^{\prime }$, and whether server downtime;10      If server downtime: store the same transition in D k times;11      else: store one transition in D;12      Sample random minibatch of transitions from D13      Update parameters θ in Q-network with Formula (5)14      Every C steps reset $\theta^{-}=\theta$15   End For16 End For

#### 4.3.3. State Control Optimization Algorithm

In Part 2 of Figure 3, state machine control is added to obtain a model-assisted joint optimization method. The state machine can record the good performance state. Based on the optimized model and the state machine, the joint optimization method can realize the continuous optimization of the server to improve global optimization.

After the first part of the algorithm is executed, some better-performing configurations are obtained, as well as a trained and tuned model. These contents will be used to guide subsequent state-controlled neighborhood searches. As depicted in Figure 5, the two experience pools that continue to update come from the data of the training process. There are three states: steady state, random state, and low-performance state. In the steady state, it uses the model trained above to select an action. In the random state, it will find the past n high-performance experiences based on memory and then perform neighbor searching based on Hamming Distance. In the low-performance state, it will find the first n high-performance experiences from pool B and then search nearby. The process of generating a new configuration for neighbor searching is as follows: (1) Randomly generate a candidate configuration. (2) Calculate the Hamming distance between the candidate configuration and n reference configurations. In the random state, the reference configuration is the past n high-performance experiences. (3) Determine whether more than half of the n distances are lower than the set distance threshold, and if so, generate a new solution; otherwise, repeat (1). At the beginning of the algorithm, the last 5% of the first stage and the downtime configuration are filled into experience pool A, and the first 1% of the configuration is filled into B. The initial configuration is set the same as the first stage and defined as a stable state. Then, state switching is performed according to Figure 5, and each state transition is an iterative process. If the resulting new solution performance is lower than the maximum value of A, it is added to A, and if it is higher than the minimum value of B, it is added to B.

## 5. Experimental Results

### 5.1. Workload Scenario Recognition Results

In each scenario, a similar amount of data is collected, with a total of 497,640 combined data. With the help of the Pearson correlation coefficient [38] for feature screening, the features with high correlation are removed, and finally, 38 features remain in Table A1. After dividing the dataset, there are 338,396 pieces of data in the training set, 59,716 pieces of data in the validation set, and 99,528 pieces of data in the test set. As seen in Figure 6, each classifier after training is evaluated using the test set. It was found that the performance of the decision tree is the worst. Its classification ability for hard disk-intensive scenarios, idle scenarios, and CPU-intensive scenarios is too poor, resulting in low accuracy. The other three algorithms have good classification capabilities for CPU-intensive scenarios, memory-intensive scenarios, and network-intensive scenarios and can achieve close to 100% accuracy. The performance difference between these four algorithms is mainly reflected in the classification ability of hard disk-intensive scenarios and idle scenarios. According to the results, DNN has the best classification ability, with an accuracy rate of 99.7%.

### 5.2. Operating Scenario Performance Optimization Experiment

According to prior knowledge, the server does not work if some BIOS switch configuration items change. Therefore, these configuration items will be directly removed. The final configuration items are shown in Table 2, and 104 switch configuration items are reserved for the experimental STREAM scenario.

#### 5.2.1. Simulation

To verify the effectiveness of the optimization algorithm detailed in Section 3 and to find a good parameter design scheme promptly, the algorithm simulation experiment is designed before the experiment.

The BIOS on the server is configured dynamically, and the STREAM benchmark tool is run to collect simulation model training data. STREAM uses four memory operations, copy, scale, add, and triad, to test the bandwidth performance of the system memory. These four rates are used as indicators to measure the server’s performance. From the collected data, the minimum average memory rate of the server in the experiment is close to 150,000 MB/s. Then, a performance score regression model is established based on the collected data.

Experiments with different hyperparameters are conducted based on the optimization algorithm in this paper, and the parameter setting scheme is selected by comparison. The influence of some hyperparameter settings on training is shown in Figure 7. When the learning rate or discount coefficient is too large, the average return shows a downward trend. The final hyperparameter design scheme is shown in Table 3. The model is used for optimization from the initial BIOS configuration, as seen in Figure 8. Moreover, the model can quickly optimize from a low-score state to a high-score state.

#### 5.2.2. Measured Experiment

We refer to the parameter setting plan of the simulation experiment and make minor modifications to it. The actual training requires a test server, which takes a long time, thus reducing the capacity of the experience pool and the final step T=40. The performance score for the initial training state is 202,000 MB/s. The average reward is used to evaluate the effect of the algorithm training on the actual server.

In Figure 9, as the number of iterations increases, the average return obtained by the algorithm continues to increase, but the shock is greater than the simulation result. In an actual server environment, some BIOS configuration items and configuration progress cause the server to crash or work unstably. When this happens, the cumulative return of the round drops considerably, and then the reward curve exhibits a sudden drop. Additionally, the trained model is used to optimize the initial BIOS configuration in Figure 10.

The state machine control of the BIOS optimization algorithm (Part 2) is added to realize continuous optimization. Table 4 shows the comparison of this paper’s joint optimization algorithm and other traditional heuristic algorithms, in which popsize represents the number of individuals in the population, and iter represents the number of iterations. The experience configuration is provided by the cooperative server manufacturer unit, which is obtained by a large number of manually automated tests. The optimal configuration performance of the three algorithms is all higher than the empirical configuration, but the BPSO algorithm has the worst optimization effect and the highest probability of downtime. This is because BPSO, despite having a strong global search capability, becomes more random with the iterative search of the algorithm, resulting in an increased probability of generating downtime configurations. Since too many downtimes will damage the server, the BPSO optimizations are all manually terminated in about 30 iterations, leading to the worst optimization effect. Different population sizes in the genetic algorithm also affected the downtime probability and optimization effect. When the number of individuals in the population is larger, the difference between populations will be larger, and the global optimization ability will also be stronger. Therefore, the performance optimization effect of popsize = 100 is better than that of popsize = 20. Moreover, individuals with greater differences are more likely to cause downtime when performing crossover operations. In addition, both algorithms need to initialize the population, which also brings some downtime configuration due to the random initialization. Although the algorithm proposed in this paper limits the number of tuning steps, the search space is also large enough, so the performance optimization effect can be close to that of the genetic algorithm. More importantly, the algorithm is based on reinforcement learning with a learning mechanism and is not disturbed by the initialization process; thus, the performance is more stable with a smaller downtime probability. After analyzing the downtime configurations, it was found that the server being down not only depends on the BIOS configuration but also on the configuration process. For example, configuration-A and configuration-B do not cause server downtime, but adjusting to configuration-B from configuration-A can cause server downtime. However, the GA algorithm has no memory of the configuration process.

We also perform static tuning on several other benchmark software, and the results are shown in Figure 11. The performance metric of Sysbench-CPU is the 95th percentage of the prime calculation event latency. The performance metric of Fio is the IOPS (input/output operations per second) under the random-read workload. For the Sysbench-CPU benchmark, there is almost no room for improvement of the empirical configuration. The main reason is that for the performance of the core, the empirical configuration has reached the optimal solution, and it is difficult to further optimize it further. In addition, although kunpeng920 provides a lot of configuration items, only some configurations are selected for the experiment in this article. Considering that the configurations adjusted in this article are mainly related to LLC and HHA, there is little opportunity for adjusting these configurations to improve the core performance but a greater opportunity for optimization in computing and communication scenarios, such as memory. In the follow-up work, energy efficiency can be used as the tuning index for scenarios where there is little room for performance improvement. Additionally, more memory-type benchmarks can be added for fine-grained scenario classification and static tuning.

## 6. Conclusions

In this study, we design a dynamic tuning framework under unknown operating scenarios based on the BIOS system. The aim is to give full consideration to the hardware capabilities of the system. First, dynamic scenario classification is implemented on the BIOS system using real-time server monitoring data. The accuracy rate reaches 99.7%. Then, the Markov model of the BIOS control decision is established. Combining reinforcement learning with the nearest neighbor search, we propose a joint BIOS control optimization method. Simulation experiments show that the algorithm proposed in this paper can optimize the server to a higher performance state. The actual experiments on the server prove that the algorithm performs well in dynamic situations, and in the STREAM scenario, it achieves good memory rate performance optimization (43.7% higher than the initial training configuration state and 10.3% higher than the experience configuration state). Compared with the genetic algorithm, the obtained optimal performance is close, and the probability of downtime during optimization is smaller. Static configuration optimization also shows that the performance in partial load scenarios is not sensitive to the configuration in this article, such as sysbench-CPU, and energy efficiency may be a good tuning indicator. Given this situation, the static optimization algorithm in this paper can still work. In summary, the dynamic tuning framework makes full use of system hardware functions and shows its effectiveness.

## Figures and Tables

**Figure 1 sensors-22-06730-f001:**
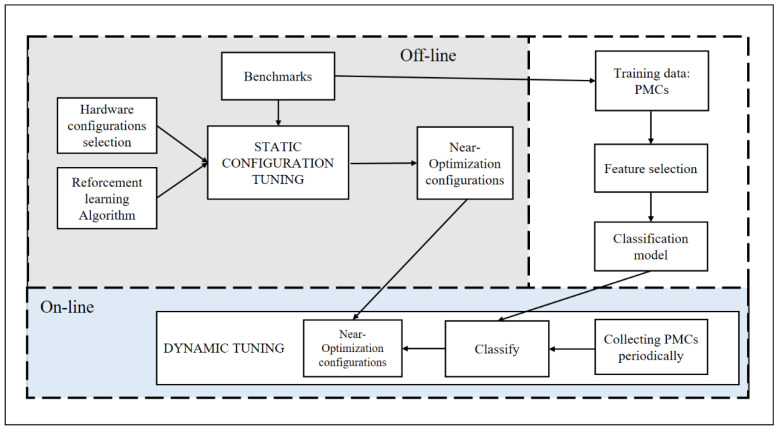
The dynamic tuning framework and workflow.

**Figure 2 sensors-22-06730-f002:**
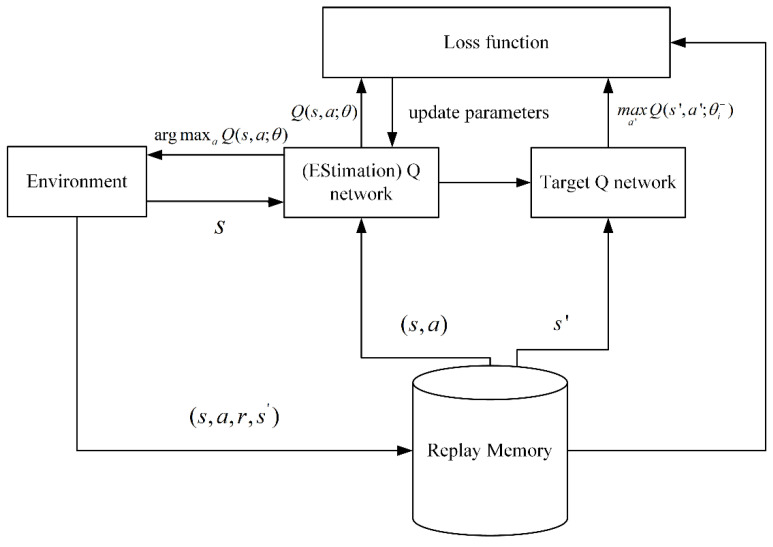
Principle of DQN.

**Figure 3 sensors-22-06730-f003:**
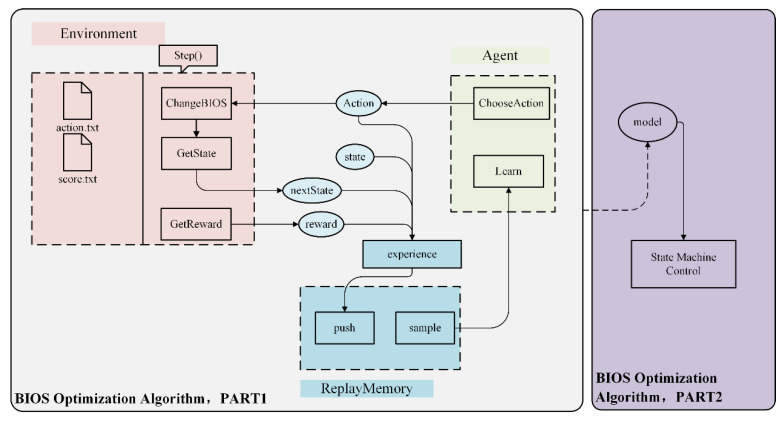
Structure of the BIOS optimization algorithm.

**Figure 4 sensors-22-06730-f004:**
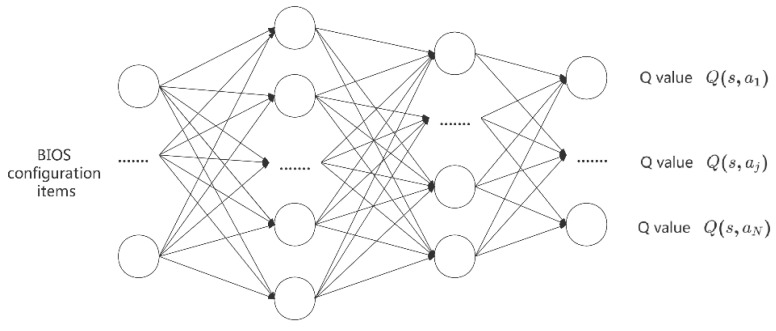
Schematic illustration of the q-network.

**Figure 5 sensors-22-06730-f005:**
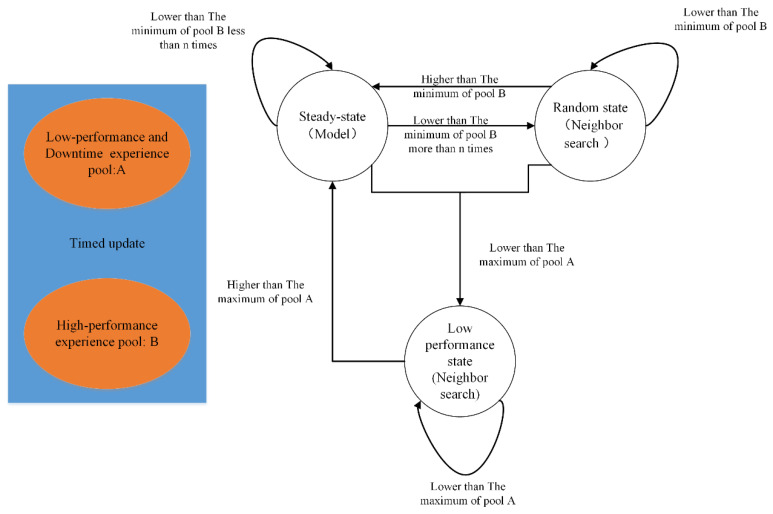
State machine process in optimization.

**Figure 6 sensors-22-06730-f006:**
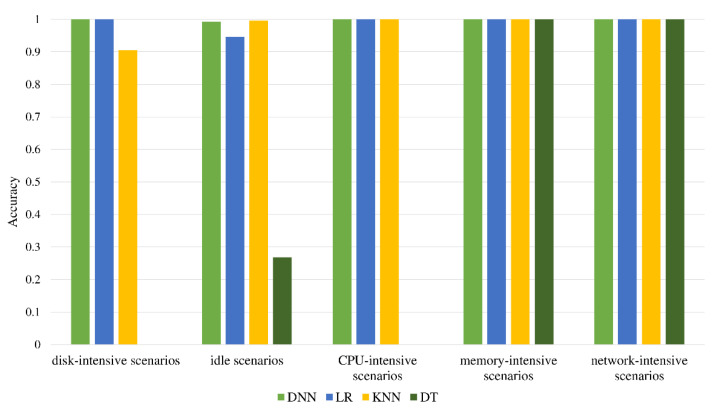
Comparison of classification accuracy.

**Figure 7 sensors-22-06730-f007:**
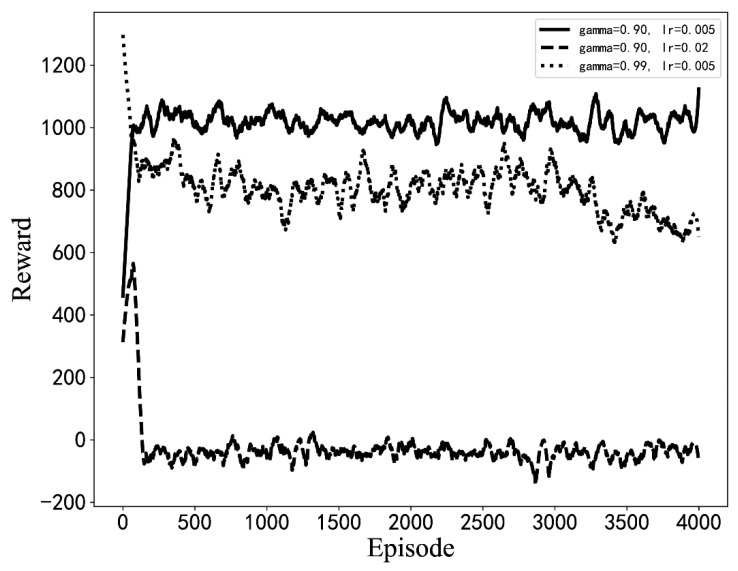
The average reward curves of the training process for different hyperparameter settings. (The results of simulation experiments).

**Figure 8 sensors-22-06730-f008:**
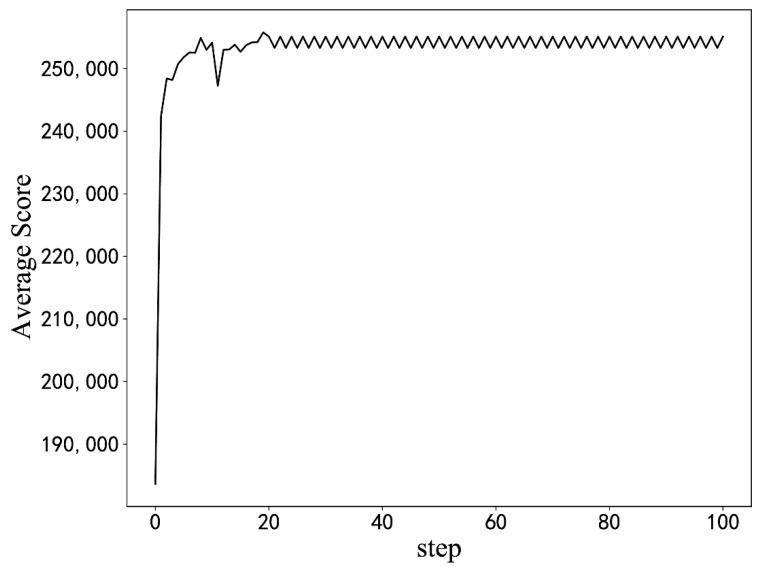
Optimization result from the initial configuration. (The results of simulation experiments).

**Figure 9 sensors-22-06730-f009:**
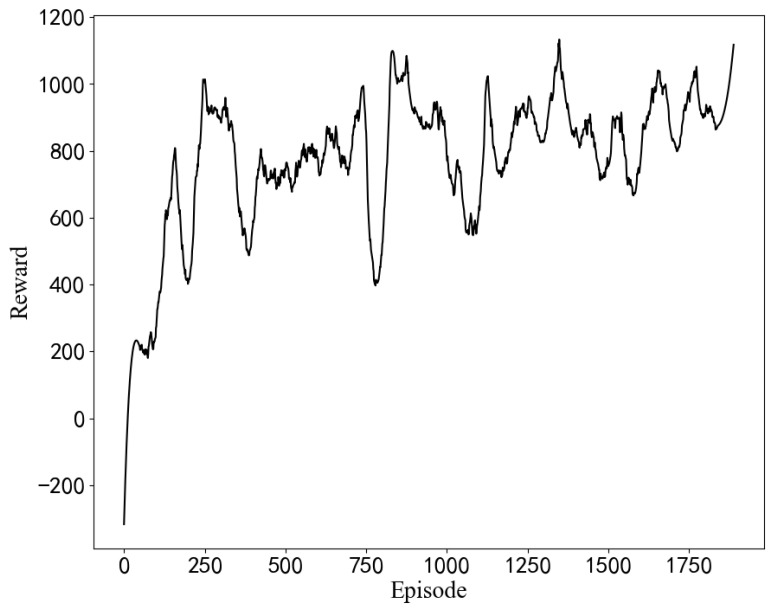
The average reward curves of the training process in the STREAM scenario. (The results of the measured experiments).

**Figure 10 sensors-22-06730-f010:**
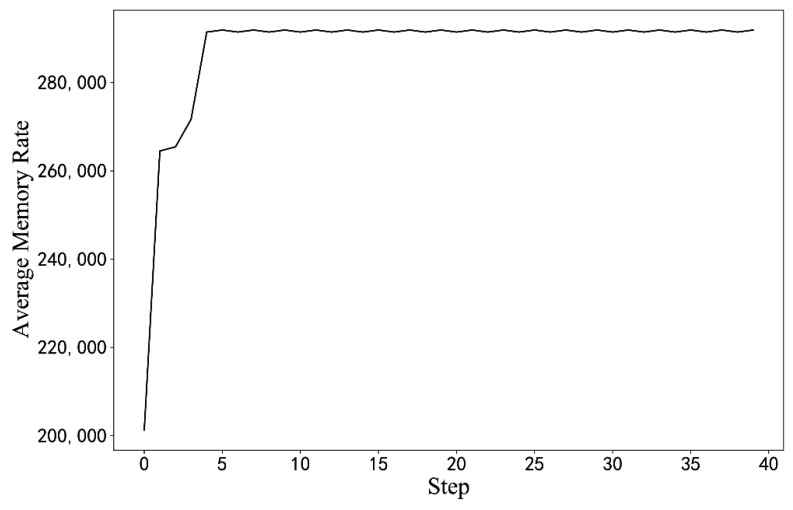
Optimization results from the initial configuration. (The results of the measured experiments).

**Figure 11 sensors-22-06730-f011:**
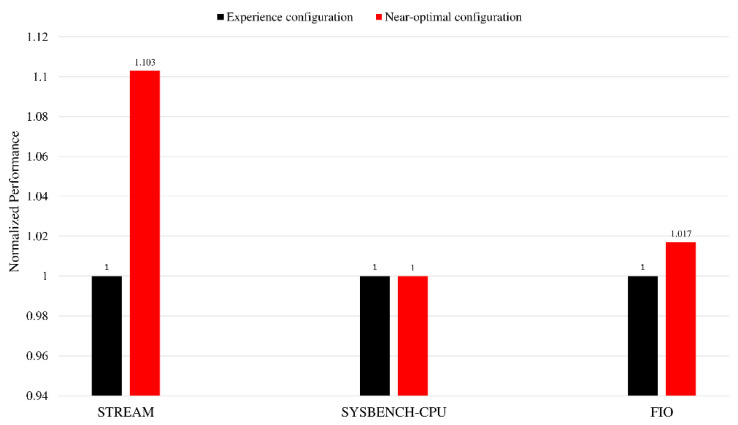
Normalized performance of experience configuration and near-optimal configuration obtained by static offline tuning.

**Table 1 sensors-22-06730-t001:** Scenario simulation tools.

Scenario Type	Benchmarking Tools
I/O-intensive scenario	FIO, IOZone
Network-intensive scenario	iPerf3
CPU-intensive scenario	Sysbench-CPU
Memory-intensive scenario	STREAM
Idle scenario	None

**Table 2 sensors-22-06730-t002:** Configuration register [9].

Register Name	Number of Configuration Bits Selected for Optimization
L3T_STATIC_CTRL	8
L3T_DYNAMIC_CTRL	16
L3T_DYNAMIC_AUCTRL0	8
L3T_DYNAMIC_AUCTRL1	21
L3T_PREFECTH	8
HHA_DIR_CTRL	15
HHA_FUNC_DIS	17
HHA_TOTEMNUM	11

**Table 3 sensors-22-06730-t003:** List of hyperparameters and their values.

Hyperparameters	Value
minibatch size	32
replay memory size	10,000
discount factor *γ*	0.90
learning rate *α*	0.005
$\mathrm{initial}\;\mathrm{exploration}\;\varepsilon_{max}$	1
$\mathrm{exploration}\;\mathrm{decay}\;\varepsilon_{decay}$	0.995
$\mathrm{final}\;\mathrm{exploration}\;\varepsilon_{min}$	0.1
final exploration frame *M*	2000
final step *T*	100

**Table 4 sensors-22-06730-t004:** Comparison of different algorithm optimization results for the STREAM scenario.

Methods	Main Parameter Settings	Best Performance Score (Average Memory Rate MB/s)	Server Downtime Probability during the Experiment Process	
Prior Knowledge		263,000		

Joint BIOS optimization algorithm using DQN (including the training process)	Iter = 50	290,400	4.1%	
Genetic algorithm	Iter = 100, popsize = 20	281,000	8.1%	
	Iter = 100, popsize = 100	289,700	20.3%	
Binary particle swarm algorithm	Popsize = 100	276,000	39.8%	

## Data Availability

Not applicable.

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
