# Peer review of "BIOS-Based Server Intelligent Optimization"

_sensors, 2022, doi:10.3390/s22186730_

Round 1

Reviewer 1 Report

The paper tackles the problem of dynamically configuring the BIOS parameters in servers, such that the servers perform optimally according to the changes in server workload. The decisions as to what parameters to change are made by machine-learned models. Previous studies configure either OS/application-level parameters or a few BIOS parameters, but the proposed study considers a larger number of BIOS parameters. In general, the paper is well written and easy to understand, and the reviewer suggests that the paper be accepted with minor changes, as follows.

(1) p4, Table 1: Can multiple scenarios in the table occur simultaneously? If they happen, is it possible that configuration changes for different scenarios conflict with each other? For example, parameter A has to be lowered according to the CPU-intensive scenario, whereas the same parameter A has to be increased according to the network-intensive scenario.

(2) p4, Section 4.1, lines 174-175, “actions are defined as the flip of N BIOS configuration items”: Can all configuration items be flipped between two values (e.g., 0 or 1)? What if a configuration item can take multiple values, such as a range of integers?

(3) p5, Section 4.1, lines 179-181: “the optional action can only be to flip a single configuration item. In this way, the configuration optimization problem can be abstracted into multistep tuning rather than reaching the best configuration in one step”: Can we always reach the best configuration eventually if we continue to flip one item at a time or we may possibly arrive at a sub-optimal configuration?

(4) p3, Figure 1: _Classify_ model --> _Classification_ model

(5) p11, Figure 7 and p12, Figure 8: Please increase the font size of the figures. They are currently too small to read.

Author Response

Dear Reviewer:

Thanks very much for taking your time to review this manuscript. Please see the attachment.

Thanks again!

Reviewer 2 Report

The authors proposed a BIOS optimization algorithm using deep Q-network. The neighborhood search method of state machine control is added improve the continuous optimization capabilities of the DQN learning model. The results look encouraging and motivating. But there are still some contents, which need be revised in order to meet the requirements of publish. A number of concerns listed as follows:

1. The abstract should be rewritten to reflect the significance of the proposed work. The current abstract shows a lot of background information.

2. In the abstract section, I would suggest that the author should provide to the point and quantitative advantages of the proposed method.

3. The main contributions of this paper should be further summarized and clearly demonstrated.

4. The method/approach in the context of the proposed work should be written in detail.

5. To explore Comparative results with existing approaches/methods relating to the proposed work.

6. Some new references should be added to improve the reviews the literatures. For example, 10.1109/JSTARS.2021.3059451; 10.1109/TR.2022.3180273; 10.3390/agriculture12060793 and so on and so on.

7.  In page 2, at Line 94, “To the best of our knowledge”à” For the best of our knowledge”,…..

8. The authors need to interpret the meanings of the variables.

9. Please add the contents of Author Contributions, Institutional Review Board Statement, Informed Consent Statement, Data Availability Statement and Conflicts of Interest.

Author Response

(The authors gave the same response as above.)

Round 2

Reviewer 2 Report

I have appreciated the deep revision of the contents and the present form of this manuscript. All my previous concerns have been accurately addressed. I think that this paper can be accepted.